# Generalization Bounds in the Predict-then-Optimize Framework

**Othman El Balghiti**
Rayens Capital
Chicago, IL 60606
oe2161@columbia.edu

**Adam N. Elmachtoub**
Columbia University
New York, NY 10027
adam@ieor.columbia.edu

**Paul Grigas**
University of California, Berkeley
Berkeley, CA 94720
pgrigas@berkeley.edu

**Ambuj Tewari**
University of Michigan
Ann Arbor, MI 48109
tewaria@umich.edu

## Abstract

The predict-then-optimize framework is fundamental in many practical settings: predict the unknown parameters of an optimization problem, and then solve the problem using the predicted values of the parameters. A natural loss function in this environment is to consider the cost of the decisions induced by the predicted parameters, in contrast to the prediction error of the parameters. This loss function was recently introduced [7] and christened Smart Predict-then-Optimize (SPO) loss. Since the SPO loss is nonconvex and noncontinuous, standard results for deriving generalization bounds do not apply. In this work, we provide an assortment of generalization bounds for the SPO loss function. In particular, we derive bounds based on the Natarajan dimension that, in the case of a polyhedral feasible region, scale at most logarithmically in the number of extreme points, but, in the case of a general convex set, have poor dependence on the dimension. By exploiting the structure of the SPO loss function and an additional strong convexity assumption on the feasible region, we can dramatically improve the dependence on the dimension via an analysis and corresponding bounds that are akin to the margin guarantees in classification problems.

## 1 Introduction

A common application of machine learning is to *predict-then-optimize*, i.e., predict unknown parameters of an optimization problem and then solve the optimization problem using the predictions. For instance, consider a navigation task that requires solving a shortest path problem. The key input into this problem are the travel times on each edge, typically called edge costs. Although the exact costs are not known at the time the problem is solved, the edge costs are predicted using a machine learning model trained on historical data consisting of features (time of day, weather, etc.) and edge costs (collected from app data). Fundamentally, a good model induces the optimization problem to find good shortest paths, as measured by the true edge costs. In fact, recent work has been developed to consider how to solve problems in similar environments [3, 13, 6]. In particular, recent work [7] developed the Smart Predict-then-Optimize (SPO) loss function which exactly measures the quality of a prediction by the decision error, in contrast to the prediction error as measured by standard loss functions such as squared error. In this work, we seek to provide an assortment of generalization bounds for the SPO loss function.

Specifically, we shall assume that our optimization task is to minimize a linear objective over a convex feasible region. In the shortest path example, the feasible region is a polyhedron. We assume the objective cost vector is not known at the time the optimization problem is solved, but rather predicted from data. A decision is made with respect to the predicted cost vector, and the SPO loss is computed by evaluating the decision on the true cost vector and then subtracting the optimal cost assuming knowledge of the true cost vector. Unfortunately, the SPO loss is nonconvex and non-Lipschitz, and therefore proving generalization bounds is not immediate.

Our results consider two cases, depending on whether the feasible region is a polyhedron or a strongly convex body. In all cases, we achieve a dependency of $\frac{1}{\sqrt{n}}$ up to logarithmic terms, where $n$ is the number of samples. In the polyhedral case, our generalization bound is formed by considering the Rademacher complexity of the class obtained by compositing the SPO loss with our predict-then-optimize models. This in turn can be bounded by a term on the order of square root of the Natarajan dimension times the logarithm of the number of extreme points in the feasible region. Since the number of extreme points is typically exponential in the dimension, this logarithm is essential so that the bound is at most linear in the dimension. When our cost vector prediction models are restricted to linear, we show that the Natarajan dimension of the predict-then-optimize hypothesis class is simply bounded by the product of the two relevant dimensions, the feature dimension and the cost vector dimension, of the linear hypothesis class. Using this polyhedral approach, we show that a generalization bound is possible for any convex set by looking at a covering of the feasible region, although the dependency on the dimension is at least linear.

Fortunately, we show that when the feasible region is strongly convex, tighter generalization bounds can be obtained using margin-based methods. The proof relies on constructing an upper bound on the SPO-loss function and showing it is Lipschitz. Our margin based bounds have no explicit dependence on dimensions of input features and of cost vectors. It is expressed as a function of the multivariate Rademacher complexity of the vector-valued hypothesis class being used. We show that for suitably constrained linear hypothesis classes, we get a much improved dependence on problem dimensions. Since the SPO loss generalizes the 0-1 multiclass loss from multiclass classification (see Example 3), our work can be seen as extending classic Natarajan-dimension based [20, Ch. 29] and margin-based generalization bounds [14] to the predict-then-optimize framework.

We note that one can generally construct an instance of a multiclass classification problem from an instance of an SPO problem by considering the "label" of each observed cost vector to be the corresponding optimal solution which is without loss of generality an extreme point of the feasible set of solutions. The number of classes in the resulting multiclass problem is the number of extreme points of the feasible set. It is therefore important to use those generalization bounds from the multiclass classification literature that are not too large in the number of classes. For data-independent worst-case bounds, the dependency is at best square root in the number of classes [9, 4]. In contrast, we provide data-independent bounds that grow only logarithmically in the number of extreme points. Using data-dependent (margin-based) approaches, [15, 16] successfully decreased this complexity to logarithm in the number of classes.

However, the reduction of an SPO problem to multiclass classification throws away potentially important information, namely the numerical values of the cost vectors. Our margin-based approach removes any explicit dependency on the number of classes by exploiting the structure of the SPO loss. In Section 4, we make an important assumption that the feasible set is strongly convex (which necessarily implies that the number of extreme points is *infinite*) and also heavily uses the structure of the SPO loss via the construction of the $\gamma$-margin SPO loss. This refined analysis allows us to circumvent a naive bound that depends on the infinite number of classes, which would be vacuous.

Even though we construct a Lipschitz upper bound on SPO loss in a general norm setting (Theorem 3), our margin bounds (Theorem 4) are stated in the $\ell_2$ norm setting. This is because the most general contraction type lemma for vector valued Lipschitz functions we know of only works for the $\ell_2$-norm [17]. Similar results are available in the infinity norm setting [3] but our understanding of general norms appears limited at present. Our work will hopefully provide the motivation to develop contraction inequalities for vector valued Lipschitz functions in a general norm setting.

## 2 Predict-then-optimize framework and preliminaries

We now describe the predict-then-optimize framework which is central to many applications of optimization in practice. Specifically, we assume that there is a nominal optimization problem of interest which models our downstream decision-making task. Furthermore, we assume that the nominal problem has a linear objective and that the decision variable $w \in \mathbb{R}^d$ and feasible region $S \subseteq \mathbb{R}^d$ are well-defined and known with certainty. However, the cost vector of the objective, $c \in \mathbb{R}^d$, is not observed directly, and rather an associated feature vector $x \in \mathbb{R}^p$ is observed. Let $\mathcal{D}$ be the underlying joint distribution of $(x, c)$ and let $\mathcal{D}_x$ be the conditional distribution of $c$ given $x$. Then the goal for the decision maker is to solve

$$\min_{w \in S} \mathbb{E}_{c \sim \mathcal{D}_x}[c^T w | x] \;=\; \min_{w \in S} \mathbb{E}_{c \sim \mathcal{D}_x}[c|x]^T w \tag{1}$$

The predict-then-optimize framework relies on using a prediction/estimate for $\mathbb{E}_{c \sim \mathcal{D}_x}[c|x]$, which we denote by $\hat{c}$, and solving the deterministic version of the optimization problem based on $\hat{c}$. We define $P(\hat{c})$ to be the optimization task with objective cost vector $\hat{c}$, namely

$$P(\hat{c}): \quad \min_w \quad \hat{c}^T w \\ \text{s.t.} \quad w \in S. \tag{2}$$

We assume $S \subseteq \mathbb{R}^d$ is a nonempty, compact, and convex set representing the feasible region. We let $w^*(\cdot) : \mathbb{R}^d \to S$ denote any *oracle* for solving $P(\cdot)$. That is, $w^*(\cdot)$ is a fixed deterministic mapping such that $w^*(c) \in \arg\min_{w \in S} \left\{ c^T w \right\}$ for all $c \in \mathbb{R}^d$. For instance, if (2) corresponds to a linear, conic, or even a particular combinatorial or mixed-integer optimization problem (in which case $S$ can be implicitly described as a convex set), then a commercial optimization solver or a specialized algorithm suffices for $w^*(c)$.

In this framework, we assume that predictions are made from a model that is learned on a training data set. Specifically, the sample training data $(x_1, c_1), \ldots, (x_n, c_n)$ is drawn i.i.d. from the joint distribution $\mathcal{D}$, where $x_i \in \mathcal{X}$ is a feature vector representing auxiliary information associated with the cost vector $c_i$. We denote by $\mathcal{H}$ our hypothesis class of cost vector prediction models, thus for a function $f \in \mathcal{H}$, we have that $f : \mathcal{X} \to \mathbb{R}^d$. Most approaches for learning a model $f \in \mathcal{H}$ from the training data are based on specifying a loss function that quantifies the error in making prediction $\hat{c}$ when the realized (true) cost vector is actually $c$. Following prior work [7], our primary loss function of interest is the "smart predict-then-optimize" loss function that directly takes the nominal optimization problem $P(\cdot)$ into account when measuring errors in predictions. Namely, we consider the SPO loss function (relative to the optimization oracle $w^*(\cdot)$) defined by:

$$\ell_{\mathrm{SPO}}(\hat{c}, c) := c^T (w^*(\hat{c}) - w^*(c)) \,,$$

where $\hat{c} \in \mathbb{R}^d$ is the predicted cost vector and $c \in \mathcal{C} \subseteq \mathbb{R}^d$ is the true realized cost vector. Notice that $\ell_{\mathrm{SPO}}(\hat{c}, c)$ exactly measures the excess cost incurred when making a suboptimal decision due to an imprecise cost vector prediction. Also, note that the SPO loss is non-negative and bounded above by $\omega_S(\mathcal{C})$ for all $\hat{c} \in \mathbb{R}^d$ and $c \in \mathcal{C}$ where $\omega_S(\mathcal{C})$ is a diameter-like quantity that we will define shortly. Let us now present several examples to illustrate the applicability and generality of the SPO loss function and framework.

*Example* 1. In the shortest path problem, the feature vector $x$ may include features such as weather and time information that may be used to predict the cost vector $c$ representing the travel times along each edge of the network. In this case, the network is assumed to be given (e.g., the road network of a city) and the feasible region $S$ is a network flow polytope that represents flow conservation and capacity constraints on the underlying network.

*Example* 2. In portfolio optimization, the returns of potential investments can depend on many features which typically include historical returns, news, economic factors, social media, and others. We presume that these auxiliary features may be used to predict the vector of returns $r$ of $d$ different assets, but that the covariance matrix of the asset returns does not depend on the auxiliary features. Here we are interested in maximizing returns, so we let the cost vector $c$ be defined by $c = -\tilde{r}$ where $\tilde{r} = r - r_{\mathrm{RF}} e$, $r$ represents the vector of asset returns, $r_{RF}$ is the risk-free rate, and $e$ is the vector of all ones. If $\Sigma \in \mathbb{R}^{d \times d}$ denotes the (positive semidefinite) covariance matrix of the asset returns and $\gamma \geq 0$ is a desired bound on the overall variance (risk level) of the portfolio, then we may define the feasible region by $S := \{w : w^T \Sigma w \leq \gamma, e^T w \leq 1, w \geq 0\}$.

*Example* 3. Our setting also captures multi-class (and binary) classification by the following characterization: $S$ is the $d$-dimensional simplex where $d$ is the number of classes, and $\mathcal{C} = \{-e_i | i = 1, \ldots, d\}$ where $e_i$ is the $i^{th}$ unit vector in $\mathbb{R}^d$. It is easy to see that each vertex of the simplex corresponds to a label, and correct/incorrect prediction has a loss of 0/1.

As pointed out before [7], the SPO loss function is generally non-convex, may even be discontinuous, and is in fact a strict generalization of the 0-1 loss function in binary classification. Thus, optimizing the SPO loss via empirical risk minimization may be intractable even when $\mathcal{H}$ is a linear hypothesis class. To circumvent these difficulties, one approach is to optimize a convex surrogate loss [7]. Our focus is on deriving generalization bounds that hold uniformly over the class $\mathcal{H}$, and thus are valid for *any* training approach, including using a surrogate or other loss function within the framework of empirical risk minimization. Notice that a generalization bound for the SPO loss directly translates to an upper bound guarantee for problem (1) that holds "on average" over the distribution.

**Useful notation.**   We will make use of a generic given norm $\| \cdot \|$ on $w \in \mathbb{R}^d$, as well as the $\ell_q$-norm denoted by $\| \cdot \|_q$ for $q \in [1, \infty]$. For the given norm $\| \cdot \|$ on $\mathbb{R}^d$, $\| \cdot \|_*$ denotes the dual norm defined by $\|c\|_* := \max_{w:\|w\| \le 1} c^T w$. Let $B(\bar{w}, r) := \{w : \|w - \bar{w}\| \le r\}$ denote the ball of radius $r$ centered at $\bar{w}$, and we analogously define $B_q(\bar{w}, r)$ for the $\ell_q$-norm and $B_*(c, r)$ for the dual norm. For a set $S \subseteq \mathbb{R}^d$, we define the size of $S$ in the norm $\| \cdot \|$ by $\rho(S) := \sup_{w \in S} \|w\|$. We analogously define $\rho_q(\cdot)$ for the $\ell_q$-norm and $\rho_*(\cdot)$ for the dual norm. We define the "linear optimization gap" of $S$ with respect to $c$ by $\omega_S(c) := \max_{w \in S} \{c^T w\} - \min_{w \in S} \{c^T w\}$, and for a set $\mathcal{C} \subseteq \mathbb{R}^d$ we slightly abuse notation by defining $\omega_S(\mathcal{C}) := \sup_{c \in \mathcal{C}} \omega_S(c)$. Define $w^*(\mathcal{H}) := \{x \mapsto w^*(f(x)) : f \in \mathcal{H}\}$.

**Rademacher complexity.**   Let us now briefly review the notion of Rademacher complexity and its application in our framework. Recall that $\mathcal{H}$ is a hypothesis class of functions mapping from the feature space $\mathcal{X}$ to $\mathbb{R}^d$. Given a fixed sample $(x_1, c_1)...(x_n, c_n)$, we define the empirical risk with respect to the SPO loss of a function $f \in \mathcal{H}$ as

$$\hat{R}_{\text{SPO}}(f) = \frac{1}{n} \sum_{i=1}^{n} \ell_{\text{SPO}}(f(x_i), c_i) ,$$

and the expected risk as $R_{\text{SPO}}(f) = \mathbb{E}_{(x,c) \sim \mathcal{D}}[\ell_{\text{SPO}}(f(x), c)]$. We also define the empirical Rademacher complexity of $\mathcal{H}$ with respect to the SPO loss, i.e., the empirical Rademacher complexity of the function class obtained by composing $\ell_{\text{SPO}}$ with $\mathcal{H}$ by

$$\hat{\mathfrak{R}}_{\text{SPO}}^n(\mathcal{H}) := \mathbb{E}_{\sigma} \left[ \sup_{f \in \mathcal{H}} \frac{1}{n} \sum_{i=1}^{n} \sigma_i \ell_{\text{SPO}}(f(x_i), c_i) \right] ,$$

where $\sigma_i$ are i.i.d. Rademacher random variables for $i = 1, \ldots, n$. The expected version of the Rademacher complexity is defined as $\mathfrak{R}_{\text{SPO}}^n(\mathcal{H}) := \mathbb{E}\left[\hat{\mathfrak{R}}_{\text{SPO}}^n(\mathcal{H})\right]$ where the expectation is w.r.t. an i.i.d. sample drawn from the underlying distribution $\mathcal{D}$. The following theorem is an application of the classical generalization bounds based on Rademacher complexity due to [1] to our setting.

**Theorem 1** (Bartlett and Mendelson [1]). *Let $\mathcal{H}$ be a family of functions mapping from $\mathcal{X}$ to $\mathbb{R}^d$. Then, for any $\delta > 0$, with probability at least $1 - \delta$ over an i.i.d. sample drawn from the distribution $\mathcal{D}$, each of the following holds for all $f \in \mathcal{H}$*

$$R_{\text{SPO}}(f) \le \hat{R}_{\text{SPO}}(f) + 2\mathfrak{R}_{\text{SPO}}^n(\mathcal{H}) + \omega_S(\mathcal{C})\sqrt{\frac{\log(1/\delta)}{2n}} , \text{ and}$$

$$R_{\text{SPO}}(f) \le \hat{R}_{\text{SPO}}(f) + 2\hat{\mathfrak{R}}_{\text{SPO}}^n(\mathcal{H}) + 3\omega_S(\mathcal{C})\sqrt{\frac{\log(2/\delta)}{2n}} .$$

## 3   Combinatorial dimension based generalization bounds

In this section, we consider the case where $S$ is a polyhedron and derive generalization bounds based on bounding the Rademacher complexity of the SPO loss and applying Theorem 1. Since $S$ is polyhedral, the optimal solution of (2) can be found by considering only the finite set of extreme points of $S$, which we denote by the set $\mathfrak{S}$. Since the number of extreme points may be exponential

in $d$, our goal is to provide bounds that are logarithmic in $|\mathfrak{S}|$. At the end of the section, we extend our analysis to any compact and convex feasible region $S$ by extending the polyhedral analysis with a covering number argument.

In order to derive a bound on the Rademacher complexity, we will critically rely on the notion of the Natarajan dimension [19], which is an extension of the VC-dimension to the multiclass classification setting and is defined in our setting as follows.

**Definition 1** (Natarajan dimension). *Suppose that $S$ is a polyhedron and $\mathfrak{S}$ is the set of its extreme points. Let $\mathcal{F} \subseteq \mathfrak{S}^{\mathcal{X}}$ be a hypothesis space of functions mapping from $\mathcal{X}$ to $\mathfrak{S}$, and let $\mathbb{X} \subseteq \mathcal{X}$ be given. We say that $\mathcal{F}$ N-shatters $\mathbb{X}$ if there exists $g_1, g_2 \in \mathcal{F}$ such that*

- *$g_1(x) \neq g_2(x)$ for all $x \in \mathbb{X}$*

- *For all $T \subseteq \mathbb{X}$, there exists $g \in \mathcal{F}$ such that (i) for all $x \in T$, $g(x) = g_1(x)$ and (ii) for all $x \in \mathbb{X} \backslash T$, $g(x) = g_2(x)$.*

*The Natarajan dimension of $\mathcal{F}$, denoted $d_N(\mathcal{F})$, is the maximal cardinality of a set N-shattered by $\mathcal{F}$.*

The Natarajan dimension is a measure for the richness of a hypothesis class. In Theorem 2, we show that the Rademacher complexity for the SPO loss can be bounded as a function of the Natarajan dimension of $w^*(\mathcal{H}) := \{x \mapsto w^*(f(x)) : f \in \mathcal{H}\}$. The proof follows a classical argument and makes strong use of Massart's lemma and the Natarajan lemma.

**Theorem 2.** *Suppose that $S$ is a polyhedron and $\mathfrak{S}$ is the set of its extreme points. Let $\mathcal{H}$ be a family of functions mapping from $\mathcal{X}$ to $\mathbb{R}^d$. Then we have that*

$$\mathfrak{R}_{\mathrm{SPO}}^n(\mathcal{H}) \leq \omega_S(\mathcal{C}) \sqrt{\frac{2d_N(w^*(\mathcal{H})) \log(n|\mathfrak{S}|^2)}{n}}.$$

*Furthermore, for any $\delta > 0$, with probability at least $1 - \delta$ over an i.i.d. sample $(x_1, c_1), \ldots, (x_n, c_n)$ drawn from the distribution $\mathcal{D}$, for all $f \in \mathcal{H}$ we have*

$$R_{\mathrm{SPO}}(f) \leq \hat{R}_{\mathrm{SPO}}(f) + 2\omega_S(\mathcal{C}) \sqrt{\frac{2d_N(w^*(\mathcal{H})) \log(n|\mathfrak{S}|^2)}{n}} + \omega_S(\mathcal{C}) \sqrt{\frac{\log(1/\delta)}{2n}}.$$

Next, we show that when $\mathcal{H}$ is restricted to the linear hypothesis class $\mathcal{H}_{\mathrm{lin}} = \{x \mapsto Bx : B \in \mathbb{R}^{d \times p}\}$, then the Natarajan dimension of $w^*(\mathcal{H}_{lin})$ can be bounded by $dp$. The proof relies on translating our problem to an instance of linear multiclass prediction problem and using a result of [5].

**Corollary 1.** *Suppose that $S$ is a polyhedron and $\mathfrak{S}$ is the set of its extreme points. Let $\mathcal{H}_{lin}$ be the hypothesis class of all linear functions, i.e., $\mathcal{H}_{\mathrm{lin}} = \{x \mapsto Bx : B \in \mathbb{R}^{d \times p}\}$. Then we have*

$$d_N(w^*(\mathcal{H}_{\mathrm{lin}})) \leq dp.$$

*Furthermore, for any $\delta > 0$, with probability at least $1 - \delta$ over an i.i.d. sample $(x_1, c_1), \ldots, (x_n, c_n)$ drawn from the distribution $\mathcal{D}$, for all $f \in \mathcal{H}_{\mathrm{lin}}$ we have*

$$R_{\mathrm{SPO}}(f) \leq \hat{R}_{\mathrm{SPO}}(f) + 2\omega_S(\mathcal{C}) \sqrt{\frac{2dp \log(n|\mathfrak{S}|^2)}{n}} + \omega_S(\mathcal{C}) \sqrt{\frac{\log(1/\delta)}{2n}}.$$

Next, we will build off the previous results to prove generalization bounds in the case where $S$ is a general compact convex set. The arguments we made earlier made extensive use of the extreme points of the polyhedron. Nevertheless, this combinatorial argument can be modified in order to derive similar results for general $S$. The approach is to approximate $S$ by a grid of points corresponding to the smallest cardinality $\epsilon$-covering of $S$. To optimize over these grid of points, we first find the optimal solution in $S$ and then round to the nearest point in the grid. Both the grid representation and the rounding procedure can fortunately both be handled by similar arguments made in Theorems 2 and Corollary 1, yielding a generalization bound below.

**Corollary 2.** *Let $S$ be any compact and convex set, and let $\mathcal{H}_{\mathrm{lin}}$ be the hypothesis class of all linear functions. Then, for any $\delta > 0$, with probability at least $1 - \delta$ over an i.i.d. sample $(x_1, c_1), \ldots, (x_n, c_n)$ drawn from the distribution $\mathcal{D}$, for all $f \in \mathcal{H}_{\mathrm{lin}}$ we have*

$$R_{\mathrm{SPO}}(f) \leq \hat{R}_{\mathrm{SPO}}(f) + 4d\omega_S(\mathcal{C}) \sqrt{\frac{2p \log(2n\rho_2(S)d)}{n}} + 3\omega_S(\mathcal{C}) \sqrt{\frac{\log(2/\delta)}{2n}} + O\left(\frac{1}{n}\right).$$

Although the dependence on the sample size $n$ in the above bound is favorable, the dependence on the number of features $p$ and the dimension of the feasible region $d$ is relatively weak. Given that the proofs of Corollary 2 and Theorem 2 are purely combinatorial and hold for worst-case distributions, this is not surprising. In the next section, we demonstrate how to exploit the structure of the SPO loss function and additional convexity properties of $S$ in order to develop improved bounds.

## 4   Margin-based generalization bounds under strong convexity

In this section, we develop improved generalization bounds for the SPO loss function under the additional assumption that the feasible region $S$ is strongly convex. Our developments are akin to and in fact are a strict generalization of the margin guarantees for binary classification based on Rademacher complexity developed in [14]. We adopt the definition of strongly convex sets presented in [10, 8], which is reviewed in Definition 2 below. Recall that $\| \cdot \|$ is a generic given norm on $\mathbb{R}^d$ and $B(\bar{w}, r) := \{w : \|w - \bar{w}\| \leq r\}$ denotes the ball of radius $r$ centered at $\bar{w}$.

**Definition 2.** *We say that a convex set $S \subseteq \mathbb{R}^d$ is $\mu$-strongly convex with respect to the norm $\| \cdot \|$ if, for any $w_1, w_2 \in S$ and for any $\lambda \in [0, 1]$, it holds that:*

$$B \left( \lambda w_1 + (1 - \lambda) w_2, \left( \tfrac{\mu}{2} \right) \lambda (1 - \lambda) \|w_1 - w_2\|^2 \right) \subseteq S \ .$$

Informally, Definition 2 says that, for every convex combination of points in $S$, a ball of appropriate radius also lies in $S$. Several examples of strongly convex sets are presented in [10, 8], including $\ell_q$ and Schatten $\ell_q$ balls for $q \in (1, 2]$, certain group norm balls, and generally any level set of a smooth and strongly convex function.

Our analysis herein relies on the following Proposition, which strengthens the first-order general optimality condition for differentiable convex optimization problems under the additional assumption of strong convexity. Proposition 1 may be of independent interest and, to the best of our knowledge, has not appeared previously in the literature.

**Proposition 1.** *Let $S \subseteq \mathbb{R}^d$ be a non-empty $\mu$-strongly convex set and let $F(\cdot) : \mathbb{R}^d \to \mathbb{R}$ be a convex and differentiable function. Consider the convex optimization problem:*

$$\begin{aligned} \min_{w} \quad & F(w) \\ \text{s.t.} \quad & w \in S \ . \end{aligned} \tag{3}$$

*Then, $\bar{w} \in S$ is an optimal solution of (3) if and only if:*

$$\nabla F(\bar{w})^T (w - \bar{w}) \geq \left( \tfrac{\mu}{2} \right) \|\nabla F(\bar{w})\|_* \|w - \bar{w}\|^2 \ \text{for all } w \in S \ . \tag{4}$$

In fact, we prove a slightly more general version of the proposition where the function $F$ need only be defined on an open set containing $S$. In the case of linear optimization with $F(w) = \hat{c}^T w$, the inequality (4) implies that $w^*(\hat{c})$ is the unique optimal solution of $P(\hat{c})$ whenever $\hat{c} \neq 0$ and $\mu > 0$. Hence, in the context of the SPO loss function with a strongly convex feasible region, $\|\hat{c}\|_*$ provides a degree of "confidence" regarding the decision $w^*(\hat{c})$ implied by the cost vector prediction $\hat{c}$. This intuition motivates us to define the "$\gamma$-margin SPO loss", which places a greater penalty on cost vector predictions near 0.

**Definition 3.** *For a fixed parameter $\gamma > 0$, given a cost vector prediction $\hat{c}$ and a realized cost vector $c$, the $\gamma$-margin SPO loss $\ell_{\mathrm{SPO}}^\gamma(\hat{c}, c)$ is defined as:*

$$\ell_{\mathrm{SPO}}^\gamma(\hat{c}, c) := \begin{cases} \ell_{\mathrm{SPO}}(\hat{c}, c) & \text{if } \|\hat{c}\|_* > \gamma \\ \left( \frac{\|\hat{c}\|_*}{\gamma} \right) \ell_{\mathrm{SPO}}(\hat{c}, c) + \left( 1 - \frac{\|\hat{c}\|_*}{\gamma} \right) \omega_S(c) & \text{if } \|\hat{c}\|_* \leq \gamma \end{cases}$$

Recall that, for any $\hat{c}, c \in \mathbb{R}^d$, it holds that $\ell_{\mathrm{SPO}}(\hat{c}, c) \leq \omega_S(c)$. Hence, we also have that $\ell_{\mathrm{SPO}}(\hat{c}, c) \leq \ell_{\mathrm{SPO}}^\gamma(\hat{c}, c)$, that is the $\gamma$-margin SPO loss provides an upper bound on the SPO loss. Notice that the $\gamma$-margin SPO loss interpolates between the SPO loss and the upper bound $\omega_S(c)$ whenever $\|\hat{c}\|_* \leq \gamma$. The $\gamma$-margin SPO loss also satisfies a simple monotonicity property whereby $\ell_{\mathrm{SPO}}^\gamma(\hat{c}, c) \leq \ell_{\mathrm{SPO}}^{\bar{\gamma}}(\hat{c}, c)$ for any $\hat{c}, c \in \mathbb{R}^d$ and $\bar{\gamma} \geq \gamma > 0$. We can also define a "hard $\gamma$-margin SPO loss" that simply returns the upper bound $\omega_S(c)$ whenever $\|\hat{c}\|_* \leq \gamma$.

**Definition 4.** *For a fixed parameter $\gamma \geq 0$, given a cost vector prediction $\hat{c}$ and a realized cost vector $c$, the hard $\gamma$-margin SPO loss $\bar{\ell}^{\gamma}_{\mathrm{SPO}}(\hat{c}, c)$ is defined as:*

$$\bar{\ell}^{\gamma}_{\mathrm{SPO}}(\hat{c}, c) := \begin{cases} \ell_{\mathrm{SPO}}(\hat{c}, c) & \text{if } \|\hat{c}\|_* > \gamma \\ \omega_S(c) & \text{if } \|\hat{c}\|_* \leq \gamma \end{cases}$$

It is simple to see that $\ell_{\mathrm{SPO}}(\hat{c}, c) \leq \ell^{\gamma}_{\mathrm{SPO}}(\hat{c}, c) \leq \bar{\ell}^{\gamma}_{\mathrm{SPO}}(\hat{c}, c) \leq \omega_S(c)$ for all $\hat{c}, c \in \mathbb{R}^d$ and $\gamma > 0$. Due to this additional upper bound, in all of the subsequent generalization bound results, the empirical $\gamma$-margin SPO loss can be replaced by its hard margin counterpart.

We are now ready to state a theorem concerning the Lipschitz properties of the optimization oracle $w^*(\cdot)$ and the $\gamma$-margin SPO loss, which will then be used to derive margin-based generalization bounds. Theorem 3 below first demonstrates that the optimization oracle $w^*(\cdot)$ satisfies a "Lipschitz-like" property away from zero. Subsequently, this Lipschitz-like property is a key ingredient in demonstrating that the $\gamma$-margin SPO loss is Lipschitz.

**Theorem 3.** *Suppose that feasible region $S$ is $\mu$-strongly convex with $\mu > 0$. Then, the optimization oracle $w^*(\cdot)$ satisfies the following "Lipschitz-like" property: for any $\hat{c}_1, \hat{c}_2 \in \mathbb{R}^d$, it holds that:*

$$\|w^*(\hat{c}_1) - w^*(\hat{c}_2)\| \leq \frac{1}{\mu \cdot \min\{\|\hat{c}_1\|_*, \|\hat{c}_2\|_*\}} \|\hat{c}_1 - \hat{c}_2\|_* . \tag{5}$$

*Moreover, for any fixed $c \in \mathbb{R}^d$ and $\gamma > 0$, the $\gamma$-margin SPO loss is $(5\|c\|_*/\gamma\mu)$-Lipschitz with respect to the dual norm $\|\cdot\|_*$, i.e., it holds that:*

$$|\ell^{\gamma}_{\mathrm{SPO}}(\hat{c}_1, c) - \ell^{\gamma}_{\mathrm{SPO}}(\hat{c}_2, c)| \leq \frac{5\|c\|_*}{\gamma\mu} \|\hat{c}_1 - \hat{c}_2\|_* \text{ for all } \hat{c}_1, \hat{c}_2 \in \mathbb{R}^d . \tag{6}$$

*Proof.* We present here only the proof of (5) and defer the proof of (6), which relies crucially on (5), to the supplementary materials. Let $\tau := \min\{\|\hat{c}_1\|_*, \|\hat{c}_2\|_*\}$. We assume without loss of generality that $\tau > 0$ (otherwise the right-hand side of (5) is equal to $+\infty$ by convention). Applying Proposition 1 twice yields:

$$\hat{c}_1^T(w^*(\hat{c}_2) - w^*(\hat{c}_1)) \geq \left(\tfrac{\mu}{2}\right)\tau\|w^*(\hat{c}_1) - w^*(\hat{c}_2)\|^2 ,$$

and

$$\hat{c}_2^T(w^*(\hat{c}_1) - w^*(\hat{c}_2)) \geq \left(\tfrac{\mu}{2}\right)\tau\|w^*(\hat{c}_1) - w^*(\hat{c}_2)\|^2 .$$

Adding the above two inequalities together yields:

$$\mu\tau\|w^*(\hat{c}_1) - w^*(\hat{c}_2)\|^2 \leq (\hat{c}_2 - \hat{c}_1)^T(w^*(\hat{c}_1) - w^*(\hat{c}_2)) \leq \|\hat{c}_1 - \hat{c}_2\|_*\|w^*(\hat{c}_1) - w^*(\hat{c}_2)\| ,$$

where the second inequality is Hölder's inequality. Dividing both sides of the above by $\mu\tau\|w^*(\hat{c}_1) - w^*(\hat{c}_2)\|$ yields the desired result. $\square$

**Margin-based generalization bounds.** We are now ready to present our main generalization bounds of interest in the strongly convex case. Our results are based on combining Theorem 3 with the Lipschitz vector-contraction inequality for Rademacher complexities developed in [17], as well as the results of [1]. Following [3, 17], given a fixed sample $((x_1, c_1)...(x_n, c_n))$, we define the *multivariate empirical Rademacher complexity* of $\mathcal{H}$ as

$$\hat{\mathfrak{R}}^n(\mathcal{H}) := \mathbb{E}_\sigma\left[\sup_{f \in \mathcal{H}} \frac{1}{n}\sum_{i=1}^n \sum_{j=1}^d \sigma_{ij} f_j(x_i)\right] = \mathbb{E}_{\boldsymbol{\sigma}}\left[\sup_{f \in \mathcal{H}} \frac{1}{n}\sum_{i=1}^n \boldsymbol{\sigma}_i^T f(x_i)\right] , \tag{7}$$

where $\sigma_{ij}$ are i.i.d. Rademacher random variables for $i = 1, \ldots, n$ and $j = 1, \ldots, d$, and $\boldsymbol{\sigma}_i = (\sigma_{i1}, \ldots, \sigma_{id})^T$. The expected version of the *multivariate Rademacher complexity* is defined as $\mathfrak{R}^n(\mathcal{H}) := \mathbb{E}\left[\hat{\mathfrak{R}}^n(\mathcal{H})\right]$ where the expectation is w.r.t. the i.i.d. sample drawn from the underlying distribution $\mathcal{D}$.

Let us also define the empirical $\gamma$-margin SPO loss and the empirical Rademacher complexity of $\mathcal{H}$ with respect to the $\gamma$-margin SPO loss as follows:

$$\hat{R}^{\gamma}_{\mathrm{SPO}}(f) := \frac{1}{n}\sum_{i=1}^n \ell^{\gamma}_{\mathrm{SPO}}(f(x_i), c_i) , \text{ and } \hat{\mathfrak{R}}^n_{\gamma\mathrm{SPO}}(\mathcal{H}) := \mathbb{E}_\sigma\left[\sup_{f \in \mathcal{H}} \frac{1}{n}\sum_{i=1}^n \sigma_i \ell^{\gamma}_{\mathrm{SPO}}(f(x_i), c_i)\right] ,$$

where $f \in \mathcal{H}$ on the left side above and $\sigma_i$ are i.i.d. Rademacher random variables for $i = 1, \ldots, n$.

In the following two theorems, we focus only on the case of the $\ell_2$-norm set-up, i.e., the norm on the space of $w$ variables as well as the norm on the space of cost vectors $c$ are both the $\ell_2$-norm. To the best of our knowledge, extending the vector-contraction inequality of [17] to an arbitrary norm setting (or even the case of general $\ell_q$-norms) remains an open question that would have interesting applications to our framework. Theorem 4 below presents our margin based generalization bounds for a fixed $\gamma > 0$. Recall that $\mathcal{C}$ denotes the domain of the true cost vectors $c$, $\rho_2(\mathcal{C}) = \sup_{c \in \mathcal{C}} \|c\|_2$, and $\omega_S(\mathcal{C}) := \sup_{c \in \mathcal{C}} \omega_S(c)$.

**Theorem 4.** *Suppose that feasible region $S$ is $\mu$-strongly convex with respect to the $\ell_2$-norm with $\mu > 0$, and let $\gamma > 0$ be fixed. Let $\mathcal{H}$ be a family of functions mapping from $\mathcal{X}$ to $\mathbb{R}^d$. Then, for any fixed sample $((x_1, c_1)...(x_n, c_n))$ we have that*

$$\hat{\mathfrak{R}}^n_{\gamma\mathrm{SPO}}(\mathcal{H}) \leq \frac{5\sqrt{2}\rho_2(\mathcal{C})\hat{\mathfrak{R}}^n(\mathcal{H})}{\gamma\mu} .$$

*Furthermore, for any $\delta > 0$, with probability at least $1 - \delta$ over an i.i.d. sample $\mathcal{S}_n$ drawn from the distribution $\mathcal{D}$, each of the following holds for all $f \in \mathcal{H}$*

$$R_{\mathrm{SPO}}(f) \leq \hat{R}^\gamma_{\mathrm{SPO}}(f) + \frac{10\sqrt{2}\rho_2(\mathcal{C})\mathfrak{R}^n(\mathcal{H})}{\gamma\mu} + \omega_S(\mathcal{C})\sqrt{\frac{\log(1/\delta)}{2n}} , \ and$$

$$R_{\mathrm{SPO}}(f) \leq \hat{R}^\gamma_{\mathrm{SPO}}(f) + \frac{10\sqrt{2}\rho_2(\mathcal{C})\hat{\mathfrak{R}}^n(\mathcal{H})}{\gamma\mu} + 3\omega_S(\mathcal{C})\sqrt{\frac{\log(2/\delta)}{2n}} .$$

*Proof.* The bound on $\hat{\mathfrak{R}}^n_{\gamma\mathrm{SPO}}(\mathcal{H})$ follows simply by combining Theorem 3, particularly (6), with equation (1) of [17]. The subsequent generalization bounds then simply follow since $R_{\mathrm{SPO}}(f) \leq R^\gamma_{\mathrm{SPO}}(f)$ for all $f \in \mathcal{H}$ and by applying the version of Theorem 1 for the $\gamma$-margin SPO loss. $\square$

It is often the case that the structure of the hypothesis class $\mathcal{H}$ naturally leads to a bound on $\mathfrak{R}^n(\mathcal{H})$ that can have mild, even logarithmic, dependence on dimensions $p$ and $d$. For example, let us consider the general setting of a constrained linear function class, namely $\mathcal{H} = \mathcal{H}_\mathcal{B} := \{f : f(x) = Bx$ for some $B \in \mathbb{R}^{d \times p}, B \in \mathcal{B}\}$, where $\mathcal{B} \subseteq \mathbb{R}^{d \times p}$. In Section A.2.4 of the supplementary materials, we derive a result that extends Theorem 3 of [12] to multivariate Rademacher complexity and provides a convenient way to bound $\mathfrak{R}^n(\mathcal{H}_\mathcal{B})$ in the case when $\mathcal{B}$ corresponds to the level set of a strongly convex function. When $\mathcal{B} = \{B : \|B\|_F \leq \beta\}$ (where $\|B\|_F$ denotes the Frobenius norm of $B$) this result implies that $\mathfrak{R}^n(\mathcal{H}_\mathcal{B}) \leq \frac{\rho_2(\mathcal{X})\beta\sqrt{2d}}{\sqrt{n}}$, and when $\mathcal{B} = \{B : \|B\|_1 \leq \beta\}$ (where $\|B\|_1$ denotes the $\ell_1$-norm of the vectorized matrix $B$) this result implies that $\mathfrak{R}^n(\mathcal{H}_\mathcal{B}) \leq \frac{\rho_\infty(\mathcal{X})\beta\sqrt{6\log(pd)}}{\sqrt{n}}$. Note the absence of any explicit dependence on $p$ in the first bound and only logarithmic dependence on $p, d$ in the second. We discuss the details of these and additional examples, including the "group-lasso" norm, in Section A.2.4.

Theorem 4 may also be extended to bounds that hold uniformly over all values of $\gamma \in (0, \bar{\gamma}]$, where $\bar{\gamma} > 0$ is a fixed parameter. This extension is presented below in Theorem 5.

**Theorem 5.** *Suppose that feasible region $S$ is $\mu$-strongly convex with respect to the $\ell_2$-norm with $\mu > 0$, and let $\bar{\gamma} > 0$ be fixed. Let $\mathcal{H}$ be a family of functions mapping from $\mathcal{X}$ to $\mathbb{R}^d$. Then, for any $\delta > 0$, with probability at least $1 - \delta$ over an i.i.d. sample drawn from the distribution $\mathcal{D}$, each of the following holds for all $f \in \mathcal{H}$ and for all $\gamma \in (0, \bar{\gamma}]$*

$$R_{\mathrm{SPO}}(f) \leq \hat{R}^\gamma_{\mathrm{SPO}}(f) + \frac{20\sqrt{2}\rho_2(\mathcal{C})\mathfrak{R}^n(\mathcal{H})}{\gamma\mu} + \omega_S(\mathcal{C}) \left( \sqrt{\frac{\log(\log_2(2\bar{\gamma}/\gamma))}{n}} + \sqrt{\frac{\log(2/\delta)}{2n}} \right) , \ and$$

$$R_{\mathrm{SPO}}(f) \leq \hat{R}^\gamma_{\mathrm{SPO}}(f) + \frac{20\sqrt{2}\rho_2(\mathcal{C})\hat{\mathfrak{R}}^n(\mathcal{H})}{\gamma\mu} + \omega_S(\mathcal{C}) \left( \sqrt{\frac{\log(\log_2(2\bar{\gamma}/\gamma))}{n}} + 3\sqrt{\frac{\log(4/\delta)}{2n}} \right) .$$

Note that a natural choice for $\bar{\gamma}$ in Theorem 5 is $\bar{\gamma} \leftarrow \sup_{f \in \mathcal{H}, x \in \mathcal{X}} \|f(x)\|_2$, presuming that one can bound this quantity based on the properties of $\mathcal{H}$ and $\mathcal{X}$. Example 4 below discusses how Theorems 4 and 5 relate to known results in binary classification.

*Example* 4. In [7], it is shown that the SPO loss corresponds exactly to the 0-1 loss in binary classification when $d = 1$, $S = [-1/2, +1/2]$, and $\mathcal{C} = \{-1, +1\}$. In this case, using our notation, the margin value of a prediction $\hat{c}$ is $c\hat{c}$. It is also easily seen that $\omega_S(\mathcal{C}) = \rho_2(\mathcal{C}) = 1$, the $\gamma$-margin SPO loss corresponds exactly to the margin loss (or ramp loss) that interpolates between 1 and 0 when $c\hat{c} \in [0, \gamma]$, and the hard $\gamma$-margin SPO loss corresponds exactly to the margin loss that returns 1 when $c\hat{c} \leq \gamma$ and 0 otherwise. Furthermore, note that the interval $S = [-\frac{1}{2}, +\frac{1}{2}]$ is 2-strongly convex [8]. Thus, except for some worse absolute constants, Theorems 4 and 5 generalize the well-known results on margin guarantees based on Rademacher complexity for binary classification [14].

As in the case of binary classification, the utility of Theorems 4 and 5 is strengthened when the underlying distribution $\mathcal{D}$ has a "favorable margin property." Namely, the bounds in Theorems 4 and 5 can be much stronger than those of Corollary 2 when the distribution $\mathcal{D}$ and the sample are such that there exists a relatively large value of $\gamma$ such that the empirical $\gamma$-margin SPO loss is small. One is thus motivated to choose the value of $\gamma$ in a data-driven way so that, given a prediction function $\hat{f}$ trained on the data $\mathcal{S}_n$, the upper bound on $\hat{R}_{\mathrm{SPO}}(\hat{f})$ is minimized. Since Theroem 5 is a uniform result over $\gamma \in (0, \bar{\gamma}]$, this data-driven procedure for choosing $\gamma$ is indeed valid.

## 5 Conclusions and Future Directions

Our work extends learning theory, as developed for binary and multiclass classification, to predict-then-optimize problems in two very significant directions: (i) obtaining worst-case generalization bounds using combinatorial parameters that measure the capacity of function classes, and (ii) exploiting special structure in data by deriving margin-based generalization bounds that scale more gracefully w.r.t. problem dimensions. It also motivates several interesting avenues for future work. Beyond the margin theory, other aspects of the problem that lead to improvements over worst case rates should be studied. In this respect, developing a theory of local Rademacher complexity for predict-then-optimize problems would be a promising approach. It will be good to use minimax constructions to provide matching lower bounds for our upper bounds. Extending the margin theory for strongly convex sets, where the SPO loss is ill-behaved only near 0, to polyhedral sets, where it can be much more ill-behaved, is a challenging but fascinating direction. Developing a theory of surrogate losses, especially convex ones, that are calibrated w.r.t. the non-convex SPO loss will also be extremely important. Finally, the assumption that the optimization objective is linear could be relaxed to include non-linear objectives.

**Acknowledgments**

OB thanks Rayens Capital for their support. AE acknowledges the support of NSF via grant CMMI-1763000. PG acknowledges the support of NSF Awards CCF-1755705 and CMMI-1762744. AT acknowledges the support of NSF via CAREER grant IIS-1452099 and of a Sloan Research Fellowship.

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
