[Supplementary Material · predict-then-optimize-supp-camera.pdf]

# A  Supplementary Materials

## A.1  Proofs for Section 3

### A.1.1  Proof of Theorem 2

*Proof.* The proof is along the lines of Corollary 3.8 in [18]. Fix a sample of data $\mathcal{S}_n = (\mathbb{X}, \mathbb{C}) \in (\mathcal{X}, \mathcal{C})^n$, where $\mathbb{X} = (x_1, \ldots, x_n)$ and $\mathbb{C} = (c_1, \ldots, c_n)$. Let $\mathfrak{F}_{|\mathbb{X}} := \{(w^*(f(x_1)), \ldots, w^*(f(x_n))) : f \in \mathcal{H}\}$. From the definition of empirical Rademacher complexity, we have that

$$
\begin{aligned}
\hat{\mathfrak{R}}_{\mathrm{SPO}}^n(\mathcal{H}) &= \mathbb{E}_\sigma \left[ \sup_{f \in \mathcal{H}} \frac{1}{n} \sum_{i=1}^n \sigma_i \ell_{\mathrm{SPO}}(f(x_i), c_i) \right] \\
&= \mathbb{E}_\sigma \left[ \sup_{f \in \mathcal{H}} \frac{1}{n} \sum_{i=1}^n \sigma_i c_i^T (w^*(f(x_i)) - w^*(c_i)) \right] \\
&= \mathbb{E}_\sigma \left[ \sup_{(w_1, \ldots, w_n) \in \mathfrak{F}_{|\mathbb{X}}} \frac{1}{n} \sum_{i=1}^n \sigma_i c_i^T (w_i - w^*(c_i)) \right] \\
&\leq \omega_S(\mathcal{C}) \sqrt{\frac{2 \log |\mathfrak{F}_{|\mathbb{X}}|}{n}} \\
&\leq \omega_S(\mathcal{C}) \sqrt{\frac{2 d_N(w^*(\mathcal{H})) \log(n |\mathfrak{S}|^2)}{n}}
\end{aligned}
$$

where the first inequality is directly due to Massart's lemma and the definition of $\omega_S(\mathcal{C})$ and the second inequality follows from the Natarajan Lemma (see Lemma 29.4 in [20]). The bound for the expected version of the Rademacher complexity follows immediately from the bound on the empirical Rademacher complexity. Applying this bound with Theorem 1 concludes the proof. $\qquad\square$

### A.1.2  Proof of Corollary 1

*Proof.* We will prove that $w^*(\mathcal{H}_{\mathrm{lin}})$ is an instance of a linear multiclass predictor for a particular class-sensitive feature mapping $\Psi$. Recall that $|\mathfrak{S}|$ is the number of extreme points of $S$. In our application of linear multiclass predictors, let $\Psi : \mathcal{X} \times \{1, \ldots, |\mathfrak{S}|\} \mapsto \mathbb{R}^{d \times p}$ be a function that takes a feature vector an extreme point and maps it to a matrix and let

$$
\mathcal{H}_\Psi = \{x \mapsto \underset{i \in \{1, \ldots, |\mathfrak{S}|\}}{\mathrm{argmax}} \ \langle B, \Psi(x, i) \rangle : B \in \mathbb{R}^{d \times p}\}.
$$

We will show that, for $\Psi(x, i) = w_i x^T$, we have that $w^*(\mathcal{H}_{\mathrm{lin}}) \subseteq \mathcal{H}_\Psi$. Consider any $f \in \mathcal{H}_{\mathrm{lin}}$ and the associated matrix $B_f$. Then

$$
\begin{aligned}
w^*(B_f x) &\in \underset{w \in S}{\mathrm{argmin}} (B_f x)^T w \\
&= \underset{i \in \{1, \ldots, |\mathfrak{S}|\}}{\mathrm{argmax}} \ -(B_f x)^T w_i \\
&= \underset{i \in \{1, \ldots, |\mathfrak{S}|\}}{\mathrm{argmax}} \ -\mathrm{Tr}\left((B_f x)^T w_i\right) \\
&= \underset{i \in \{1, \ldots, |\mathfrak{S}|\}}{\mathrm{argmax}} \ -\mathrm{Tr}\left(B_f^T w_i x^T\right) \\
&= \underset{i \in \{1, \ldots, |\mathfrak{S}|\}}{\mathrm{argmax}} \ \langle -B_f, w_i x^T \rangle.
\end{aligned}
$$

Thus, it is clear that for $\Psi(x, i) = w_i x^T$, choosing the function in $\mathcal{H}_\Psi$ corresponding to $-B_f$ yields exactly the function $f$. Therefore $w^*(\mathcal{H}_{\mathrm{lin}}) \subseteq \mathcal{H}_\Psi$. Theorem 7 in [5] shows that $d_N(\mathcal{H}_\Psi) \leq dp$. Since $w^*(\mathcal{H}_{\mathrm{lin}}) \subseteq \mathcal{H}_\Psi$, then $d_N(w^*(\mathcal{H}_{\mathrm{lin}})) \leq dp$. Combining this bound on the Natarajan dimension with Theorem 2 concludes the proof. $\qquad\square$

### A.1.3 Proof of Corollary 2

*Proof.* Consider the smallest cardinality $\epsilon$-covering of the feasible region $S$ by Euclidean balls of radius $\epsilon$. From Example 27.1 in [20], the number of balls needed is at most $\left(\frac{2\rho_2(S)\sqrt{d}}{\epsilon}\right)^d$. Let the set $\tilde{\mathfrak{S}}$ denote the centers of the balls from the smallest cardinality covering. Then it immediately follows that

$$|\tilde{\mathfrak{S}}| \le \left(\frac{2\rho_2(S)\sqrt{d}}{\epsilon}\right)^d. \tag{8}$$

Finally, let the function $\tilde{w}: S \mapsto \{1,\ldots,|\tilde{\mathfrak{S}}|\}$ be the function that takes a feasible solution in $S$ and maps it to the closest point in in $\tilde{\mathfrak{S}}$.

We can bound the empirical Rademacher complexity by

$$
\begin{aligned}
\hat{\mathfrak{R}}_{\text{SPO}}^n(\mathcal{H}) &= \mathbb{E}_\sigma\left[\sup_{f\in\mathcal{H}}\frac{1}{n}\sum_{i=1}^n \sigma_i \ell_{\text{SPO}}(f(x_i), c_i)\right]\\
&= \mathbb{E}_\sigma\left[\sup_{f\in\mathcal{H}}\frac{1}{n}\sum_{i=1}^n \sigma_i c_i^T\left(w^*(f(x_i)) - w^*(c_i)\right)\right]\\
&= \mathbb{E}_\sigma\left[\sup_{f\in\mathcal{H}}\frac{1}{n}\sum_{i=1}^n \sigma_i c_i^T\left[w^*(f(x_i)) - \tilde{w}(w^*(f(x_i))) + \tilde{w}(w^*(f(x_i))) - w^*(c_i)\right]\right]\\
&\le \mathbb{E}_\sigma\left[\sup_{f\in\mathcal{H}}\frac{1}{n}\sum_{i=1}^n \sigma_i c_i^T\left[w^*(f(x_i)) - \tilde{w}(w^*(f(x_i)))\right]\right] + \mathbb{E}_\sigma\left[\sup_{f\in\mathcal{H}}\frac{1}{n}\sum_{i=1}^n \sigma_i c_i^T\left[\tilde{w}(w^*(f(x_i))) - w^*(c_i)\right]\right]\\
&\le 2\epsilon\rho_2(\mathcal{C}) + \mathbb{E}_\sigma\left[\sup_{f\in\mathcal{H}}\frac{1}{n}\sum_{i=1}^n \sigma_i c_i^T\left[\tilde{w}(w^*(f(x_i))) - w^*(c_i)\right]\right]\\
&\le 2\epsilon\rho_2(\mathcal{C}) + (\omega_S(\mathcal{C}) + 2\epsilon\rho_2(\mathcal{C}))\sqrt{\frac{2d_N(\tilde{w}(w^*(\mathcal{H})))\log(n|\tilde{\mathfrak{S}}|^2)}{n}} \tag{9}
\end{aligned}
$$

The first inequality follows from the triangle inequality. The second inequality follows from the fact that $w^*(f(x_i))$ and $\tilde{w}(w^*(f(x_i)))$ are at most $2\epsilon$ away by the definition of $\tilde{w}$. In the worst case, the difference is in the direction of $c_i$, and $c_i^T\left[w^*(f(x_i)) - \tilde{w}(w^*(f(x_i)))\right] \le 2||c_i|| \le \rho_2(\mathcal{C})$. The third inequality follows from the same exact argument as that in Theorem 2, with the additional observation that the maximum value of $c_i^T\left[\tilde{w}(w^*(f(x_i))) - w^*(c_i)\right]$ is $\omega_S(\mathcal{C}) + 2\epsilon\rho_2(\mathcal{C})$ using a similar reasoning as in the second inequality. Thus, all that remains is to bound $d_N(\tilde{w}(w^*(\mathcal{H})))$. To do this, we first observe that $d_N(w^*(\mathcal{H})) \le dp$, where the proof follows exactly that of Corollary 1 but we now have an infinite number of labels, i.e., each point in $S$. Finally, we observe that

$$d_N(\tilde{w}(w^*(\mathcal{H}))) \le d_N(w^*(\mathcal{H})) \le dp \tag{10}$$

since $\tilde{w}$ is simply a deterministic function, and thus the number of dichotomies (labelings) that can be generated by $\tilde{w}(w^*(\mathcal{H}))$ is at most that of $w^*(\mathcal{H})$.

Now setting $\epsilon = \frac{1}{n}$, and combining Eq. (8), Eq. (9), and Eq. (10) yields

$$
\begin{aligned}
\hat{\mathfrak{R}}_{\text{SPO}}^n(\mathcal{H}) &\le \frac{2\rho_2(\mathcal{C})}{n}\left(1 + \sqrt{\frac{2dp\log(n(2n\rho_2(S)\sqrt{d})^{2d})}{n}}\right) + \omega_S(\mathcal{C})\sqrt{\frac{2dp\log(n(2n\rho_2(S)\sqrt{d})^{2d})}{n}}\\
&\le \frac{2\rho_2(\mathcal{C})}{n}\left(1 + 2d\sqrt{\frac{2p\log(2n\rho_2(S)d)}{n}}\right) + 2d\omega_S(\mathcal{C})\sqrt{\frac{2p\log(2n\rho_2(S)d)}{n}}. \tag{11}
\end{aligned}
$$

Finally, combining Eq. (11) with Theorem 2 yields

$$R_{\text{SPO}}(f) \le \hat{R}_{\text{SPO}}(f) + 4d\omega_S(\mathcal{C})\sqrt{\frac{2p\log(2n\rho_2(S)d)}{n}} + 3\omega_S(\mathcal{C})\sqrt{\frac{\log(2/\delta)}{2n}} + O\left(\frac{1}{n}\right).$$

$\square$

## A.2 Proofs for Section 4

### A.2.1 Proof of Proposition 1

*Proof.* The well-known optimality condition for differentiable convex optimization problems (see Proposition 1.1.8 of [2]) states that $\bar{w} \in S$ is an optimal solution of (3) if and only if:

$$\nabla F(\bar{w})^T (w - \bar{w}) \geq 0 \ \text{ for all } w \in S \ . \tag{12}$$

Let us now demonstrate that the conditions (4) and (12) are equivalent when $S$ is $\mu$-strongly convex.

Clearly, (4) implies (12). Now suppose that (12) holds and let $w \in S$ be arbitrary. Define $\hat{w}(\lambda) := \lambda w + (1 - \lambda)\bar{w}$ and $r(\lambda) := \left(\frac{\mu}{2}\right) \lambda(1 - \lambda)\|w - \bar{w}\|^2$ for $\lambda \in [0, 1]$. By the $\mu$-strong convexity of $S$, we have that $B(\hat{w}(\lambda), r(\lambda)) \subseteq S$. Hence, applying (12) inside $B(\hat{w}(\lambda), r(\lambda))$ yields:

$$\nabla F(\bar{w})^T (\tilde{w} - \bar{w}) \geq 0 \ \text{ for all } \tilde{w} \in B(\hat{w}(\lambda), r(\lambda)) \ .$$

Clearly the above condition is equivalent to:

$$-\nabla F(\bar{w})^T \bar{w} \ \geq \ \max_{\tilde{w} \in B(\hat{w}(\lambda), r(\lambda))} \left\{ -\nabla F(\bar{w})^T \tilde{w} \right\} \ = \ -\nabla F(\bar{w})^T \hat{w}(\lambda) + r(\lambda)\|\nabla F(\bar{w})\|_* \ ,$$

where the equality above follows from the definition of the dual norm $\| \cdot \|_*$. Rearranging the above and using $\hat{w}(\lambda) - \bar{w} = \lambda(w - \bar{w})$ as well as the definition of $r(\lambda)$ yields:

$$\lambda \nabla F(\bar{w})^T (w - \bar{w}) \ \geq \ \left(\frac{\mu}{2}\right) \lambda(1 - \lambda)\|\nabla F(\bar{w})\|_* \|w - \bar{w}\|^2 \ \text{ for all } \lambda \in [0, 1] \ .$$

Now suppose that $\lambda > 0$. Dividing the above by $\lambda$ yields:

$$\nabla F(\bar{w})^T (w - \bar{w}) \ \geq \ \left(\frac{\mu}{2}\right) (1 - \lambda)\|\nabla F(\bar{w})\|_* \|w - \bar{w}\|^2 \ \text{ for all } \lambda \in (0, 1] \ .$$

Taking the limit as $\lambda \to 0$ yields (4). $\qquad \square$

### A.2.2 Proof of Theorem 3

In this section, we complete the proof of Theorem 3 by demonstrating that (6) holds, i.e., that the $\gamma$-margin SPO loss is Lipschitz. Let us first present the following lemma that will be useful in proving (6). Recall that $B_*(c, r) = \{\hat{c} : \|\hat{c} - c\|_* \leq r\}$ is the dual norm ball centered at $c$ of radius $r$.

**Lemma 1.** *Consider the function $h_\gamma(\cdot, c) : B_*(0, \gamma) \to \mathbb{R}$ defined by $h_\gamma(\hat{c}, c) := \left(\frac{\|\hat{c}\|_*}{\gamma}\right) \ell_{\text{SPO}}(\hat{c}, c)$ for all $\hat{c} \in B_*(0, \gamma)$. Then, $h_\gamma(\cdot, c)$ is Lipschitz with respect to the dual norm $\| \cdot \|_*$ with constant $\frac{1}{\gamma}\left(\frac{\|c\|_*}{\mu} + \omega_S(c)\right) \leq \frac{3\|c\|_*}{\gamma\mu}$.*

*Proof.* Let $\hat{c}_1, \hat{c}_2 \in B_*(0, \gamma)$ be given. Note that if either $\hat{c}_1 = 0$ or $\hat{c}_2 = 0$, then the result follows since $\ell_{\text{SPO}}(\cdot, c) \leq \omega_S(c)$. Now suppose without loss of generality that $0 < \|\hat{c}_1\|_* \leq \|\hat{c}_2\|_*$. Let $\Delta := |h_\gamma(\hat{c}_1, c) - h_\gamma(\hat{c}_2, c)|$. Then, we have that

$$
\begin{aligned}
\Delta \ &= \ \left| \left(\frac{\|\hat{c}_1\|_*}{\gamma}\right) \ell_{\text{SPO}}(\hat{c}_1, c) - \left(\frac{\|\hat{c}_2\|_*}{\gamma}\right) \ell_{\text{SPO}}(\hat{c}_2, c) \right| \\
&= \ \left| \left(\frac{\|\hat{c}_1\|_*}{\gamma}\right) \ell_{\text{SPO}}(\hat{c}_1, c) - \left(\frac{\|\hat{c}_1\|_*}{\gamma}\right) \ell_{\text{SPO}}(\hat{c}_2, c) + \left(\frac{\|\hat{c}_1\|_*}{\gamma}\right) \ell_{\text{SPO}}(\hat{c}_2, c) - \left(\frac{\|\hat{c}_2\|_*}{\gamma}\right) \ell_{\text{SPO}}(\hat{c}_2, c) \right| \\
&= \ \left| \left(\frac{\|\hat{c}_1\|_*}{\gamma}\right) [\ell_{\text{SPO}}(\hat{c}_1, c) - \ell_{\text{SPO}}(\hat{c}_2, c)] + \left(\frac{\ell_{\text{SPO}}(\hat{c}_2, c)}{\gamma}\right) [\|\hat{c}_1\|_* - \|\hat{c}_2\|_*] \right| \\
&\leq \ \left(\frac{\|\hat{c}_1\|_*}{\gamma}\right) \left(\frac{\|c\|_*}{\mu\|\hat{c}_1\|_*}\right) \|\hat{c}_1 - \hat{c}_2\|_* + \left(\frac{\omega_S(c)}{\gamma}\right) \|\hat{c}_1 - \hat{c}_2\|_* \\
&= \ \frac{1}{\gamma}\left(\frac{\|c\|_*}{\mu} + \omega_S(c)\right) \|\hat{c}_1 - \hat{c}_2\|_* \ ,
\end{aligned}
$$

where the inequality above uses (5) and the reverse triangle inequality. Now we also claim that, due to the strong convexity of $S$, we have that $\omega_S(c) \leq \frac{2\|c\|_*}{\mu}$. When $c = 0$, this inequality is trivial.

Otherwise, let us apply (5) with $\hat{c}_1 \leftarrow -c$ and $\hat{c}_2 \leftarrow c$, which yields:

$$
\begin{aligned}
\omega_S(c) &= \max_{w \in S}\left\{c^T w\right\} - \min_{w \in S}\left\{c^T w\right\} \\
&= c^T(w^*(\hat{c}_1) - w^*(\hat{c}_2)) \\
&\leq \|c\|_* \|w^*(\hat{c}_1) - w^*(\hat{c}_2)\| \\
&\leq \frac{2\|c\|_*^2}{\mu\|c\|_*} = \frac{2\|c\|_*}{\mu} \; .
\end{aligned}
$$

$\square$

**Remainder of the proof of Theorem 3.** We are now ready to complete the proof of (6). Without loss of generality, we consider three cases: *(i)* $\|\hat{c}_1\|_* \leq \gamma$ and $\|\hat{c}_2\|_* \leq \gamma$, *(ii)* $\|\hat{c}_1\|_* > \gamma$ and $\|\hat{c}_2\|_* > \gamma$, and *(iii)* $\|\hat{c}_1\|_* \leq \gamma$ and $\|\hat{c}_2\|_* > \gamma$.

Let us first consider case *(i)*, i.e., we have that $\hat{c}_1, \hat{c}_2 \in B_*(0, \gamma)$. For any $\hat{c} \in B_*(0, \gamma)$, we have that

$$
\ell^\gamma_{\mathrm{SPO}}(\hat{c}, c) = \left(\frac{\|\hat{c}\|_*}{\gamma}\right)\ell_{\mathrm{SPO}}(\hat{c}, c) + \left(1 - \frac{\|\hat{c}\|_*}{\gamma}\right)\omega_S(c) \; .
$$

Hence, on the ball $B_*(0, \gamma)$, the function $\ell^\gamma_{\mathrm{SPO}}(\cdot, c)$ decomposes as the sum of three functions. By Lemma 1, we have that the function in the first term of the right-hand side above is $\frac{3\|c\|_*}{\gamma\mu}$-Lipschitz on $B_*(0, \gamma)$. Clearly, the function in the second term is $\frac{\omega_S(c)}{\gamma}$-Lipschitz. Thus, using $\omega_S(c) \leq \frac{2\|c\|_*}{\mu}$ and adding these two Lipschitz constants together yields the desired result for case *(i)*.

Now, in case *(ii)*, we have that $\ell^\gamma_{\mathrm{SPO}}(\hat{c}_1, c) = \ell_{\mathrm{SPO}}(\hat{c}_1, c)$ and $\ell^\gamma_{\mathrm{SPO}}(\hat{c}_2, c) = \ell_{\mathrm{SPO}}(\hat{c}_2, c)$. Hence, (5) yields:

$$
|\ell^\gamma_{\mathrm{SPO}}(\hat{c}_1, c) - \ell^\gamma_{\mathrm{SPO}}(\hat{c}_2, c)| = |c^T(w^*(\hat{c}_1) - w^*(\hat{c}_2))| \leq \|c\|_*\|w^*(\hat{c}_1) - w^*(\hat{c}_2)\| \leq \frac{\|c\|_*}{\gamma\mu}\|\hat{c}_1 - \hat{c}_2\|_* \; ,
$$

and clearly $\frac{\|c\|_*}{\gamma\mu} \leq \frac{5\|c\|_*}{\gamma\mu}$.

Finally, in case *(iii)*, define $\bar{c} := \lambda\hat{c}_1 + (1-\lambda)\hat{c}_2$ where $\lambda \in (0, 1]$ is such that $\|\bar{c}\|_* = \gamma$. Then, we have that:

$$
\begin{aligned}
|\ell^\gamma_{\mathrm{SPO}}(\hat{c}_1, c) - \ell^\gamma_{\mathrm{SPO}}(\hat{c}_2, c)| &= |(\ell^\gamma_{\mathrm{SPO}}(\hat{c}_1, c) - \ell^\gamma_{\mathrm{SPO}}(\bar{c}, c)) + (\ell^\gamma_{\mathrm{SPO}}(\bar{c}, c) - \ell^\gamma_{\mathrm{SPO}}(\hat{c}_2, c))| \\
&\leq |\ell^\gamma_{\mathrm{SPO}}(\hat{c}_1, c) - \ell^\gamma_{\mathrm{SPO}}(\bar{c}, c)| + |\ell^\gamma_{\mathrm{SPO}}(\bar{c}, c) - \ell^\gamma_{\mathrm{SPO}}(\hat{c}_2, c)| \\
&\leq \frac{5\|c\|_*}{\gamma\mu}\|\hat{c}_1 - \bar{c}\| + \frac{5\|c\|_*}{\gamma\mu}\|\bar{c} - \hat{c}_2\| \\
&= \frac{5\|c\|_*}{\gamma\mu}(\|\hat{c}_1 - \bar{c}\| + \|\bar{c} - \hat{c}_2\|) \\
&= \frac{5\|c\|_*}{\gamma\mu}\|\hat{c}_1 - \hat{c}_2\| \; ,
\end{aligned}
$$

where the second inequality follows from cases *(i)* and *(ii)*, and the final equality follows since $\bar{c}$ lies on the line segment between $\hat{c}_1$ and $\hat{c}_2$, i.e., we have that $\|\hat{c}_1 - \bar{c}\| = (1-\lambda)\|\hat{c}_1 - \hat{c}_2\|$ and $\|\bar{c} - \hat{c}_2\| = \lambda\|\hat{c}_1 - \hat{c}_2\|$. $\square$

Finally, it is worth pointing out that the proofs of Lemma 1 and the remainder of the proof of Theorem 4 imply that the Lipschitz constant of $\ell^\gamma_{\mathrm{SPO}}$ can be improved slightly from $\frac{5\|c\|_*}{\gamma\mu}$ to $\frac{1}{\gamma}\left(\frac{\|c\|_*}{\mu} + 2\omega_S(c)\right)$.

### A.2.3 Proof of Theorem 5

*Proof.* We prove the first inequality only; the second inequality can be proven in an identical manner. The argument here follows closely the proof of Theorem 5.9 of [18]. Define $\epsilon := \omega_S(\mathcal{C})\sqrt{\frac{\log(2/\delta)}{2n}}$ and two sequences $\{\gamma_k\}_{k=1}^\infty$ and $\{\epsilon_k\}_{k=1}^\infty$ by

$$
\epsilon_k := \epsilon + \omega_S(\mathcal{C})\sqrt{\frac{\log(k)}{n}} \; , \text{ and } \gamma_k := \frac{\bar{\gamma}}{2^k} \; , \text{ for } k \geq 1 \; .
$$

Define the following events:

$$A_k := \left\{ \sup_{f\in\mathcal{H}} \left\{ R_{\text{SPO}}(f) - \hat{R}_{\text{SPO}}^{\gamma_k}(f) - \frac{10\sqrt{2}\rho_2(\mathcal{C})\mathfrak{R}^n(\mathcal{H})}{\gamma_k\mu} - \epsilon_k \right\} > 0 \right\} \text{ for } k \geq 1 \,, \ \tilde{A} := \bigcup_{k=1}^{\infty} A_k \,, \text{ and}$$

$$\check{A} := \left\{ \sup_{f\in\mathcal{H}, \gamma\in(0,\bar{\gamma}]} \left\{ R_{\text{SPO}}(f) - \hat{R}_{\text{SPO}}^{\gamma}(f) - \frac{20\sqrt{2}\rho_2(\mathcal{C})\mathfrak{R}^n(\mathcal{H})}{\gamma\mu} - \omega_S(\mathcal{C})\sqrt{\frac{\log(\log_2(2\bar{\gamma}/\gamma))}{n}} - \epsilon \right\} > 0 \right\} \,.$$

Let us first argue that $\check{A} \subseteq \tilde{A}$. Indeed, suppose that $\check{A}$ occurs. Then, there exists some $f \in \mathcal{H}$ and some $\gamma \in (0, \bar{\gamma}]$ such that:

$$R_{\text{SPO}}(f) - \hat{R}_{\text{SPO}}^{\gamma}(f) - \frac{20\sqrt{2}\rho_2(\mathcal{C})\mathfrak{R}^n(\mathcal{H})}{\gamma\mu} - \omega_S(\mathcal{C})\sqrt{\frac{\log(\log_2(2\bar{\gamma}/\gamma))}{n}} - \epsilon > 0 \,. \qquad (13)$$

By definition of the sequence $\{\gamma_k\}$, there exists $k \geq 1$ such that $\gamma_k \leq \gamma \leq 2\gamma_k$. Thus, $\gamma_k \leq \gamma$ implies that $\hat{R}_{\text{SPO}}^{\gamma_k}(f) \leq \hat{R}_{\text{SPO}}^{\gamma}(f)$. Moreover, $\gamma \leq 2\gamma_k$ implies that $-1/\gamma_k \geq -2/\gamma$, $k \leq \log_2(2\bar{\gamma}/\gamma)$, and thus

$$\epsilon_k = \epsilon + \omega_S(\mathcal{C})\sqrt{\frac{\log(k)}{n}} \leq \epsilon + \omega_S(\mathcal{C})\sqrt{\frac{\log(\log_2(2\bar{\gamma}/\gamma))}{n}} \,.$$

Now, combining the previous inequalities together with (13) yields:

$$R_{\text{SPO}}(f) - \hat{R}_{\text{SPO}}^{\gamma_k}(f) - \frac{10\sqrt{2}\rho_2(\mathcal{C})\mathfrak{R}^n(\mathcal{H})}{\gamma_k\mu} - \epsilon_k > 0 \,,$$

which means that the event $A_k$ and correspondingly the event $\tilde{A}$ have occurred.

Now, for each $k \geq 1$, we apply Theorem 4 using $\gamma \leftarrow \gamma_k$ and $\delta \leftarrow \exp((-2n\epsilon_k^2)/\omega_S(\mathcal{C})^2)$, which yields $\mathbb{P}(A_k) \leq \exp((-2n\epsilon_k^2)/\omega_S(\mathcal{C})^2)$. We now apply $\mathbb{P}(\check{A}) \leq \mathbb{P}(\tilde{A})$ and the union bound to obtain:

$$\begin{aligned}
\mathbb{P}(\check{A}) &\leq \sum_{k=1}^{\infty} \exp\left( -\frac{2n\epsilon_k^2}{\omega_S(\mathcal{C})^2} \right) \\
&= \sum_{k=1}^{\infty} \exp\left( -2n\left( \sqrt{\frac{\log(2/\delta)}{2n}} + \sqrt{\frac{\log(k)}{n}} \right)^2 \right) \\
&< \sum_{k=1}^{\infty} \exp\left( -(\log(2/\delta) + 2\log(k)) \right) \\
&= \frac{\delta}{2} \sum_{k=1}^{\infty} \frac{1}{k^2} = \frac{\delta}{2} \cdot \frac{\pi^2}{6} < \delta \,.
\end{aligned}$$

Thus, we have completed the proof. $\qquad\qquad\qquad\qquad\qquad\qquad\qquad\qquad\qquad\qquad\square$

### A.2.4 Bounding the multivariate Rademacher complexity for linear classes

Here we use arguments in [17, 12] to bound $\mathfrak{R}^n(\mathcal{H}_{\mathcal{B}})$ where

$$\mathcal{H}_{\mathcal{B}} = \{ f \ : \ f(x) = Bx \text{ for some } B \in \mathbb{R}^{d\times p}, B \in \mathcal{B} \}$$

is the class of linear maps with matrix $B$ constrained to lie in some set $\mathcal{B}$. The following result extends Theorem 3 of [12] to multivariate Rademacher complexity.

**Theorem 6.** *Let $\mathcal{S}$ be a closed convex set and let $F : \mathcal{S} \to \mathbb{R}$ be $\alpha$-strongly convex w.r.t. $\|\cdot\|_*$ s.t. $\inf_{B\in\mathcal{S}} F(B) = 0$. Let $X$ be such that*

$$\sup_{\boldsymbol{\sigma}\in\{\pm1\}^d} \sup_{x\in\mathcal{X}} \|\boldsymbol{\sigma}x^T\| \leq X.$$

*Define $\mathcal{B} = \{ B \in \mathcal{S} \ : \ F(B) \leq \beta_*^2 \}$. Then, we have*

$$\mathfrak{R}^n(\mathcal{H}_{\mathcal{B}}) \leq X\beta_* \sqrt{\frac{2}{\alpha n}}.$$

*Proof.* Define $\boldsymbol{\sigma}_i = (\sigma_{i1}, \ldots, \sigma_{id})^T$. Then, we have

$$\mathfrak{R}^n(\mathcal{H}_\mathcal{B}) = \mathbb{E}\left[\sup_{B \in \mathcal{B}} \frac{1}{n} \sum_{i=1}^n \sum_{j=1}^d \sigma_{ij} (Bx_i)_j\right] = \mathbb{E}\left[\sup_{B \in \mathcal{B}} \frac{1}{n} \sum_{i=1}^n \boldsymbol{\sigma}_i^T B x_i\right]$$

$$= \mathbb{E}\left[\sup_{B \in \mathcal{B}} \frac{1}{n} \sum_{i=1}^n \text{Tr}\left(\boldsymbol{\sigma}_i^T B x_i\right)\right] = \mathbb{E}\left[\sup_{B \in \mathcal{B}} \frac{1}{n} \sum_{i=1}^n \text{Tr}\left(B x_i \boldsymbol{\sigma}_i^T\right)\right]$$

$$= \mathbb{E}\left[\sup_{B \in \mathcal{B}} \text{Tr}\left(B \left(\frac{1}{n} \sum_{i=1}^n \boldsymbol{\sigma}_i x_i^T\right)^T\right)\right]$$

$$= \mathbb{E}\left[\sup_{B \in \mathcal{B}} \langle B, \frac{1}{n} \sum_{i=1}^n \boldsymbol{\sigma}_i x_i^T \rangle\right].$$

Note that the inner product between matrices $B, A \in \mathbb{R}^{d \times p}$ is defined as

$$\langle B, A \rangle = \sum_{i,j} B_{ij} A_{ij} = \text{Tr}\left(B A^T\right)$$

Now fix $x_1, \ldots, x_n$ and note that, by our assumption, we have, for all $i$,

$$\sup_{\boldsymbol{\sigma} \in \{\pm 1\}^d} \|\boldsymbol{\sigma} x_i^T\| \leq X.$$

Let $\Theta$ be the random matrix $\frac{1}{n} \sum_{i=1}^n \boldsymbol{\sigma}_i x_i^T$. Choose arbitrary $\lambda > 0$. By Fenchel's inequality,

$$\langle B, \lambda\Theta \rangle \leq \frac{F(B)}{\lambda} + \frac{F^*(\lambda\Theta)}{\lambda}$$

Since $F(B) \leq \beta_*^2$ for all $B \in \mathcal{B}$, we have

$$\sup_{B \in \mathcal{B}} \langle B, \lambda\Theta \rangle \leq \frac{\beta_*^2}{\lambda} + \frac{F^*(\lambda\Theta)}{\lambda}$$

Taking expectations (w.r.t. $\sigma_{ij}$) gives

$$\mathbb{E}[\sup_{B \in \mathcal{B}} \langle B, \lambda\Theta \rangle] \leq \frac{\beta_*^2}{\lambda} + \frac{\mathbb{E}[F^*(\lambda\Theta)]}{\lambda}$$

Now let $Z_i = \frac{\lambda}{n} \boldsymbol{\sigma}_i x_i^T$ so that $S_n = \sum_{i=1}^n Z_i = \Theta$. Note that $\|Z_i\| \leq \frac{\lambda}{n} X$. So the conditions of Lemma 4 in [12] are satisfied with $V^2 = \lambda^2 X^2/n$. That lemma gives us $\mathbb{E}[F^*(\lambda\Theta)] \leq \lambda^2 X^2/(2\alpha n)$. Plugging this above, we have

$$\mathbb{E}[\sup_{B \in \mathcal{B}} \langle B, \lambda\Theta \rangle] \leq \frac{\beta_*^2}{\lambda} + \frac{\lambda X^2}{2\alpha n}.$$

Setting $\lambda = \sqrt{\frac{2\alpha n \beta_*^2}{X^2}}$ gives

$$\mathbb{E}[\sup_{B \in \mathcal{B}} \langle B, \lambda\Theta \rangle] \leq X \beta_* \sqrt{\frac{2}{\alpha n}}$$

which completes the proof. $\qquad\qquad\qquad\qquad\qquad\qquad\qquad\qquad\qquad\qquad\qquad\square$

This theorem can be applied with many different strongly convex functions of matrices [11, Section 2.4]. We give some interesting examples below.

*Example* 5 (Bounded Frobenius norm). The most basic case is $F(B) = \frac{1}{2}\|B\|_F^2$ which is 1-strongly convex on $\mathbb{R}^{d \times p}$ w.r.t. $\|\cdot\|_F$. Note that

$$\sup_{\boldsymbol{\sigma} \in \{\pm 1\}^d} \sup_{x \in \mathcal{X}} \|\boldsymbol{\sigma} x^T\|_F = \sup_{\boldsymbol{\sigma} \in \{\pm 1\}^d} \|\boldsymbol{\sigma}\|_2 \cdot \sup_{x \in \mathcal{X}} \|x\|_2 = \sqrt{d} \sup_{x \in \mathcal{X}} \|x\|_2.$$

Therefore, if $\frac{1}{2}\|B\|_F^2 \leq \beta_*^2$ and $\sup_{x \in \mathcal{X}} \|x\|_2 \leq X_2$ we have

$$\mathfrak{R}^n(\mathcal{H}_\mathcal{B}) \leq X_2 \beta_* \sqrt{\frac{2d}{n}}.$$

*Example* 6 (Bounded $\ell_1$ norm of vectorized matrix). Another case is when $\frac{1}{2}\|B\|_1^2 \leq \beta_*^2$ where $\|B\|_q$ is $\ell_q$ norm of the vectorized matrix $B$. We set $F(B) = \frac{1}{2}\|B\|_q^2$ for $q = \frac{\log(pd)}{\log(pd)-1}$ which is $1/(3\log(pd))$-strongly convex w.r.t. $\|\cdot\|_1$ [11, Corollary 10]. Since $\|B\|_q \leq \|B\|_1$, we clearly have $F(B) \leq \beta_*^2$. Note that the dual norm is $\|\cdot\|_{p'}$ for $p' = \log(pd)$ and $\|\Theta\|_{p'} \leq 3\|\Theta\|_\infty$ for any $\Theta \in \mathbb{R}^{d\times p}$. Therefore,

$$\sup_{\boldsymbol{\sigma}\in\{\pm 1\}^d}\sup_{x\in\mathcal{X}}\|\boldsymbol{\sigma}x^T\|_{p'} \leq 3 \sup_{\boldsymbol{\sigma}\in\{\pm 1\}^d}\|\boldsymbol{\sigma}\|_\infty \cdot \sup_{x\in\mathcal{X}}\|x\|_\infty = 3\sup_{x\in\mathcal{X}}\|x\|_\infty.$$

The final conclusion is that, if $\frac{1}{2}\|B\|_1^2 \leq \beta_*^2$ and $\sup_{x\in\mathcal{X}}\|x\|_\infty \leq X_\infty$ we have

$$\mathfrak{R}^n(\mathcal{H}_\mathcal{B}) \leq X_\infty \beta_* \sqrt{\frac{6\log(pd)}{n}}.$$

*Example* 7 (Bounded group-lasso norm). In case where input dimension $p$ is large, we might want to encode prior knowledge that only a subset of the $p$ input variables are relevant for making predictions. The vectorized $\ell_1$ norm considered in the previous example encourages sparsity but does not result in shared sparsity structure over the rows of $B$. That is, it does not cause entire columns to be set to zero. In multivariate regression, the group-lasso norm [21, Section 4.3] is used to enforce such a structured from of sparsity. Define the norm

$$\|B\|_{2,q} = \left(\sum_{j=1}^{p}\|B_{\cdot j}\|_2^q\right)^{1/q}.$$

The subscripts above remind us that we first take the $\ell_2$ norms of columns $B_{\cdot j}$ and then take the $\ell_q$ norm of the $p$ resulting values. The group-lasso norm is simply $\|\cdot\|_{2,1}$. Let us consider the case when the matrices $B$ are constrained to have low group-lasso norm, i.e. $\frac{1}{2}\|B\|_{2,1}^2 \leq \beta_*^2$. We set $F(B) = \frac{1}{2}\|B\|_{2,q}^2$ for $q = \frac{\log(p)}{\log(p)-1}$ which is $1/(3\log(p))$-strongly convex w.r.t. $\|\cdot\|_{2,1}$ [11, Corollary 14]. Since $\|B\|_{2,q} \leq \|B\|_{2,1}$, we clearly have $F(B) \leq \beta_*^2$. Note that the dual norm is $\|\cdot\|_{2,p'}$ for $p' = \log(p)$ and $\|\Theta\|_{2,p'} \leq 3\|\Theta\|_{2,\infty}$ for any $\Theta \in \mathbb{R}^{d\times p}$. Therefore,

$$\begin{aligned}\sup_{\boldsymbol{\sigma}\in\{\pm 1\}^d}\sup_{x\in\mathcal{X}}\|\boldsymbol{\sigma}x^T\|_{2,p'} &\leq 3 \sup_{\boldsymbol{\sigma}\in\{\pm 1\}^d}\sup_{x\in\mathcal{X}}\|\boldsymbol{\sigma}x^T\|_{2,\infty}\\
&= 3\sup_{\boldsymbol{\sigma}\in\{\pm 1\}^d}\|\boldsymbol{\sigma}\|_2 \cdot \sup_{x\in\mathcal{X}}\|x\|_\infty\\
&\leq 3\sqrt{d}\sup_{x\in\mathcal{X}}\|x\|_\infty.\end{aligned}$$

The final conclusion is that, if $\frac{1}{2}\|B\|_{2,1}^2 \leq \beta_*^2$ and $\sup_{x\in\mathcal{X}}\|x\|_\infty \leq X_\infty$ we have

$$\mathfrak{R}^n(\mathcal{H}_\mathcal{B}) \leq X_\infty \beta_* \sqrt{\frac{6d\log(p)}{n}}.$$