[Reviews · NeurIPS 2019]

Reviewer 1



The paper proposes theoretical bounds of the predict and optimize problem, using Natarjan dimensions and margin based bound ideas. The bounds appear novel and contribute towards the analysis of the predict and optimize problem, but unfortunately this isn't my field and difficult to review the novelty and innovation of the paper. The presentation is clear, and even for someone not in the field, the main logic and flow can be followed. One suggestion is it would make it more accessible by providing one or two sentences in each section to give the overall picture of how that section fits with the overall logic and contributions of the paper. This will allow readers, even if not in the area, to get a better sense of the overall picture.

Reviewer 2



I have read the author rebuttal. This is a good work. I have nothing to add. ==== This paper studies the problem of minimizing a linear function on a compact convex set, where the loss vector is not known but needs to be predicted given a feature vector and a dataset of feature-loss pairs. The problem formulation is not new and can be found in [7]. While [7] focuses on the computational aspect, this submission provides the first (as far as I know) generalization error guarantee. The statistical learning formulation is reasonable. The first generalization error bound in Section 3 is standard. The derivation of the second generalization error bound in Section 4 is sufficiently novel and interesting to me. The presentation is outstanding; all required information is given without obvious redundancy; the technical results are well connected by the paragraphs. As the first learning theoretic analysis of a well motivated problem, the significance of this submission is obvious. In particular, there are many immediate open problems as discussed in Section 5.

Reviewer 3



(1) In the introduction, the authors claimed that their margin-based approach removes the dependency on the number of classes by exploiting the structure of the SPO loss function. However, I can not see clearly this. As far as I can see, there is a logarithmic dependency for $\ell_1$-norm constrained ball and a sqrt dependency for $\ell_2$-norm constrained ball. (2) the comparison with related work is not clear. The authors should present comparison with existing work more clearly to show how this work advance the state of the art. For example, how the results applied to multi-class classification can be compared with existing works. (3) In Theorem 1, the risks and empirical risks are used but defined in Section 4. (4) It seems that several notations are defined but not used in the main text, e.g., $B_q(\bar{w},r)$. These notations can be defined in the supplementary material if they are used there only. --------------------- After authors' response: I have read the authors' response as well as other reviewer comments. I would like to keep my original score.

Reviewer 4



This paper considers a learning framework called predict-then-optimize. The problem in this setting is that parameters which are used in making predictions are not necessarily at hand when predictions should be made (costs of taking certain roads at particular moment are needed when a route has to be planned), and should be predicted before optimizing over them (in previous example, costs in the past are known and associated with other features known also at the moment). The interesting part of the framework is, that the learning problem used in learning the costs uses a loss function over error on the decision of the optimizer (SPO loss), instead of a direct error over the learned cost. In this framework, the authors provide several generalization bounds in different settings over a linear objective function, such as when feasible region where problem is solved is either polyhedron or any compact and convex region. They further work with stronger convexity assumptions and in their framework generalize margin guarantees for binary classification, and also give two modified versions of the SPO loss. Originality: The work takes a previously-known learning framework and contributes to it by giving generalization bounds for its nonconvex and noncontinuous loss. Quality: I did not check the proofs. Clarity: The paper is well written and the progression is easy to follow, although some details could be clarified for the non-expert reader (for example R_SPO in Theorem 1) Significance: The theoretical results on the difficult SPO loss can encourage further work with it. Authors also describe avenues for continuing research in other settings where Rademacher complexities could be proven. However SPO loss has not attracted much attention in the machine learning community previously (paper introducing SPO they cite is only in arXiv and not yet published). ---------------------- Edit: I have read the author's response, and stand by my review; I think that this is a good paper. As authors have addressed my questions in their responses, I have increased the score from 7 to 8. The work is clearly original and first contribution of a kind.

[Author Response · NeurIPS 2019]

We thank all reviewers for their time and feedback. Major comments are addressed below. Minor comments (typos/stylistic changes/adding expository text) will be addressed in the final version to make it more reader friendly.

**R1:** *"difficult to review the novelty and innovation of the paper"* We understand that the paper was outside of your main area of interest. Note that SPO is a new framework that includes multiclass classification as a special case. By developing combinatorial dimension based and margin-based generalization bounds for the SPO framework, we are rebuilding the two major pillars of standard generalization theory in a more challenging, novel setting.

**R2:** Thanks for your careful reading of our paper and an accurate understanding of our contributions. We are glad you highlighted the importance of: 1) construction of generalization theory for the SPO framework for the first time, 2) a generalized margin loss function and margin-theory in the strongly convex case, and 3) opening up a new area of investigation for researchers at the intersection of Operations Research (OR) and Machine Learning (ML).

**R3:** *"the authors claimed that their margin-based approach removes the dependency on the number of classes ... However, I can not see clearly this."* We agree that a bit of context is needed to interpret our statement in the introduction. We will revise the introduction accordingly. The main point is to see that one can generally construct a multiclass classification instance from an instance of an SPO problem by considering the "label" of each observed cost vector $c_i$ to be the corresponding optimal solution $w^*(c_i)$, which is w.l.o.g. an extreme point of $S$. Thus, the "number of classes" is the number of extreme points of $S$. Note however that this reduction throws away potentially important information, namely the numerical values of the cost vectors $c_i$. (Note also that Example 1 presents a case where this reduction does not remove information, in which case $S$ is the unit simplex in $\mathbb{R}^d$ with $d$ extreme points.) Now, notice that the margin based approach of Section 4 makes an important assumption that $S$ is strongly convex (which necessarily implies that the number of extreme points/classes is *infinite*) and also heavily uses the structure of the SPO loss via the construction of the $\gamma$-margin SPO loss. This refined analysis allows us to circumvent a naive bound that depends on the infinite number of classes, which would be vacuous. In fact, the dependence on the dimension $d$ is often only mild – e.g. logarithmic or square root as you mention – and improves upon the linear dependency in the bound generated via the discretization argument presented in Corollary 2.

*"It is not clear to me whether the dependency is optimal"* In specific cases, such as multiclass classification, our bounds are comparable with the best available ones (see below). As R2 pointed out, this is the *first* work on generalization theory for SPO loss. Therefore, in the general case, optimality has not been investigated. We hope our work paves the way for deriving matching lower bounds (mentioned in open problems in Section 5).

*"The authors should present comparison with existing work more clearly to show how this work advance the state of the art"* Just to clarify, ours is the *first* work to develop generalization and margin theory for the SPO framework. At that level of generality, related work simply does not exist.

*"how the results applied to multi-class classification can be compared with existing works"* Since multiclass classification is a special case, it does make sense to compare bounds. Many multiclass approaches are theoretically compared in "Multiclass Learning Approaches: A Theoretical Comparison with Implications" by Daniely et al. Different approaches (like one-vs-all, all-pairs) use different hypothesis classes. The one which is most relevant to us is their MSVM approach that uses a single space of multiclass classifiers without reducing the problem to binary classification problems. In the linear hypothesis case with $d$ classes and $p$ features, they show that the bound of $\tilde{O}\left(\sqrt{pd/n}\right)$ is tight up to logarithmic factors. Since $d_N(w^*(\mathcal{H})) \leq dp$ in the case of linear classifiers, our result is also tight up to logarithmic factors. We will add this comparison to the final version.

**R4:** *"SPO loss has not attracted much attention in the machine learning community previously"* The general problem of understanding the impact of errors in machine learning predictions when they are fed to other decision analysis tools has been gathering attention (we cite 2 neurips papers [6, 13] and 2 OR papers [3,7]). Note that OR journals have long reviewing periods which explains why the references are to their arXiv versions. The problem will increase in importance as ML is integrated in real-world decision making pipelines.

*"Theorem 1: it is not very clear how it is related to one in Bartlett and Mendelson, if these are exactly the same or not."* Theorem 1 is indeed a minor rewriting of a landmark result of Bartlett and Mendelson in a form that is easy for us to use in our setting.

*"Some connections between this work and multi-task representation learning as in Maurer et al JMLR 2016 as $c_i$ could maybe be seen as a representation of the data".* Representation learning in the SPO framework is a great idea! Actually $c_i$'s are cost vectors and are more closely related to *labels (or outputs)* rather than *examples (or inputs)* in standard supervised learning. Representation learning in the sense of Maurer et al learns the representation of *examples* and would definitely make sense if we had several related SPO problems (e.g., shortest path for different cities) to solve. Your comment reinforces our conviction that our work has much to offer to the ML community in terms of fruitful follow-up directions!

[Meta-Review · NeurIPS 2019]

This paper provides contributions regarding generalization in the predict-then-optimize framework, that suggests to value the quality of a ML system not only wrt to some prediction loss but to a decision loss. The contribution is of quality, addressing the little studied SPO framework in ML community. In addition to the theoretical results, it would be nice if the authors could squeeze in more examples (in the line of the shortest path problem), this would contribute to an effort aiming at providing some perspective on the paper.